# Midwifery Continuity of Care in Indonesia: Initiation of Mobile Health Development Integrating Midwives’ Competency and Service Needs

**DOI:** 10.3390/ijerph192113893

**Published:** 2022-10-26

**Authors:** Ari Indra Susanti, Mohammad Ali, Asep Herry Hernawan, Fedri Ruluwedrata Rinawan, Wanda Gusdya Purnama, Indriana Widya Puspitasari, Alyxia Gita Stellata

**Affiliations:** 1Faculty of Education Science, Indonesia University of Education, Jl. Dr. Setiabudi No. 229, Isola, Bandung 40154, Indonesia; 2Department of Public Health, Faculty of Medicine, Universitas Padjadjaran, Jalan Ir. Soekarno KM. 21, Jatinangor, Sumedang 45363, Indonesia; 3Center for Health System Study and Health Workforce Education Innovation, Faculty of Medicine, Universitas Padjadjaran, Jl. Eyckman No. 38, Bandung 40161, Indonesia; 4Indonesian Society for Remote Sensing branch West Java, Gedung 2, Fakultas Perikanan dan Ilmu Kelau-tan, Universitas Padjadjaran, Jl. Ir. Soekarno KM. 21, Jatinangor, Sumedang 45363, Indonesia; 5Informatics Engineering Study Program, Faculty of Engineering, Universitas Pasundan, Jl. Dr. Setiabudi No.193, Bandung 40153, Indonesia

**Keywords:** competency needs, midwifery continuity of care training, mobile health

## Abstract

Midwives’ competence in providing continuity of care using mobile health (mHealth) applications is limited in developing countries. This study identified and explored midwives’ competency and service needs to develop mHealth in Midwifery Continuity of Care (MCOC) education and training. It used an explanatory sequential mixed method, and was conducted from August to December 2021. A cross-sectional approach was used to find the characteristics and competency scope of 373 midwives in West Java, and continued with a qualitative design through a Focus Group Discussion (FGD) of 13 midwives. Descriptive data analysis (frequency, mean, deviation standard) and qualitative data analysis (coding, sub-themes, and theme) were conducted. In terms of the midwives who participated in this study, more than half were aged ≤ 35 years (58.98%), with a working period > 10 years (56.30%), had diploma degrees (71.12%), and used smartphones on average 1–12 h/day (78.28%). Most midwives needed to develop competency in the MCOC scope, including its early detection of the risk factor of complications and treatment management. They were concerned about the purposes, benefits, and design of mHealth. In summary, midwives’ competency indicators for early detection are more needed in MCOC using mHealth. Further research is required to evaluate midwives’ competence in MCOC using mHealth.

## 1. Introduction

The World Health Organization (WHO) 2017 reported that around 810 women die every day due to complications of pregnancy or childbirth in the world [1]. In Indonesia, decreasing the Maternal Mortality Rate (MMR) is still an unfinished agenda, with 305 deaths per 100,000 live births. Meanwhile, Sustainable Development Goals (SDGs) target a reduction in MMR by 70 per 100,000 live birth in 2030 [2]. In this case, the WHO recommends that all developing countries provide midwifery services with continuity of care at the primary care level to accelerate the reduction in MMR [3]. Midwifery continuity of care (MCOC) starts from pregnancy, childbirth, and postpartum to provide satisfaction for mothers to receive midwifery care [4,5]. Maternal satisfaction with maternity care is an essential indicator of the quality of delivery services [6]. This is because the quality of maternal and infant health services is determined by the management of MCOC through a systematic problem-solving process, starting from data analysis (subjective and objective), which aims to obtain actual and potential midwifery diagnoses, problems, and needs, planning, implementation, and up to evaluation [7].

The service objectives of midwives are increasing rapidly, while the existing working conditions and abilities of midwives are challenged to meet the growing medical needs of pregnant women [8]. Midwives can be essential sources of support for childbearing women and their families [9]. A systematic review revealed that MCOC could build a trusting relationship between mothers and midwives [10]. However, the MCOC has not continued to be implemented because the lack of midwives’ experience affects inadequate skills [11]. From the mother’s side, the support from midwives is influenced by geographical reasons, such as distance and access to midwives in the community [12]. Meanwhile, on the midwives’ side, the mother–midwives’ relationship of trust can be affected by midwives’ lack of work–life balance [13]. To overcome the causes, midwives need teamwork, such as with other midwives, nurses, and doctors [14]. Artificial intelligence (AI) also can help midwives to educate mothers through a conversational agent called a chatbot [15] or a combination of a chatbot and midwives’ direct conversation [16]. Supporting midwives’ competency regarding MCOC is crucial [17], including using mHealth [16,18]. However, a recent review showed limited competencies of midwives due to the lack of experience in conducting MCOC [19].

The core MCOC competencies that midwives possess are still a top priority for preventing and dealing with the causes of maternal death [20] and are hence vital in reducing maternal and newborn mortality [21,22]. Nonetheless, midwives still face challenges in carrying out competencies because they lack confidence, impacting their performances [23]. Midwives need broader knowledge and skills to meet the comprehensive care-needs of newborns and postpartum mothers [24]. In this case, midwives require regular training to increase their professional competency [25]. Competence-based in-service training, on-the-job mentoring, up-to-date standard job aids, and recognition of high-performing midwives are recommended to improve the quality of maternity care in public health facilities [26]. Currently, the need for midwife training is still high, but the instructors and training quality are still not effective. Thus, innovative, context-based, and technology-based approaches can be alternatives to overcome this problem [27].

The advances in mobile technologies and applications drive the global transformation in health services [28]. Mobile health (mHealth) applications (apps) are readily accessible to the average user of mobile devices, despite the potential of mHealth applications to improve the availability, affordability, and effectiveness of delivering healthcare services [29]. Education focuses on three aspects: cognitive, affective, and psychomotor [30]. Those aspects should be balanced, particularly psychomotor, through training to optimize the educational results [31]. In midwifery, the terms education and training may be distinguished in their use [32]. Education refers to initial knowledge, whereas training is the extension of knowledge into practical application [33]. Training strengthens midwives’ skills, which is part of the MCOC [34]. Through mHealth, learning can be a new, challenging method of health promotion for midwives and their communities. Today, mobile systems enable connectivity to the web, which is essential in health care and plays a role in continuous learning for health professionals and health promotion for the community [35]. According to previous research, there have been several applications that midwives have used to provide mother–child health (MCH), such as e-cohort [36], Tele-CTG [37], iPosyandu [38], Continuum of Care Services (CCS) [39], and Traditional Birth Attendants (TBAs) mHealth for traditional attendance [40]. However, recent literature reviews found limited specific mHealth for midwives since most of them are focused on clients’ services [39,40,41,42]. 

In Indonesia, a midwife is the essential frontier profession that supervises community-based integrated posts at the village level, called *Pos Pelayanan Terpadu* (*Posyandu*). *Posyandu* is run by community health workers (CHWs)/cadres, who perform activities for MCH: registration, height and weight measurement, documentation and reporting, and health education. The village midwife examines pregnant women and provides immunization and contraception services while supervising CHWs in *Posyandu*. To facilitate data quality verification in *Posyandu* and MCH data monitoring, Rinawan et al. built *Posyandu* mHealth, called iPosyandu, in 2017, starting for CHWs, currently extended to midwives/*bidan*, named the iPosyandu Bidan app [16,38,43]. This app development is conducted as it is essential for fulfilling the need to increase midwives’ competencies integrated with their services based on mobile health that can facilitate midwives to develop themselves according to their profession [18]. The application menu should refer to the midwives’ competencies and service needs to ensure the quality of mHealth [44], as midwives need to be well prepared to have the capacity to change client health outcomes through the adoption of evidence-based practice. In this case, education and training that are appropriate to the context and meet the different needs of midwives must be available to support the professional growth of midwives [45]. A needs assessment was conducted to understand midwives’ education and training needs, including addressing gaps in the availability and access to quality education in general and training in particular [46]. A recent systematic literature review stated an urgent need for midwives’ competency because of existing gaps in quality competency involving education, training, and services’ provision [34]. In the last decade, the gaps solved in simulated learning can potentially benefit midwives’ skills because it impacts educational and clinical skills [47]. Nowadays, solving the gaps with technological advances is challenging [33]. Competency in using mHealth to solve those gaps is limited [48]. The development process should identify and address the competency needs of midwives in education and training [49]. This study aimed to identify and explore midwives’ competency and service needs to develop mHealth in MCOC education and training.

## 2. Materials and Methods

### 2.1. Study Design

This study used an explanatory sequential mixed method, consisting of quantitative research using a cross-sectional design and qualitative research using Focus Group Discussion (FGD) (Table 1). Quantitative research was conducted from August to November 2021, and then qualitative research was performed in December 2021. First, the quantitative analysis aimed to identify the needs of midwives for the characteristics and competency scope of MCOC using mHealth. After that, mHealth was developed according to the needs of midwives in providing MCOC using qualitative research.

### 2.2. Recruitment and Participants

Quantitative research subjects included midwife participants in West Java Province, Indonesia, using a convenience sampling technique. It resulted in 373 midwives who participated in this research. The research used a questionnaire containing the competence of midwives based on the scope of early detection and treatment of risk complications during (1) pregnancy, (2) childbirth, and (3) newborns, as well as (4) the puerperium contained in the Decree of the Minister of Health of the Republic of Indonesia No. 320 of 2020 concerning the professional standards of midwives. The risk of complications refers to risk factors that are found before complications. For example, when performing anamne-sis, a midwife can assess determinants, such as at-risk behaviors and history (the “4 toos”: too young; too old; too close in labor time; and too many children); additionally, on performing a physical examination, e.g., vital signs, head to toe, including abdomen examinations (Leopold I–IV), and then laboratory tests [50]. The questionnaire comprised 22 items. A likert scale was employed with answer choices (ordinal values): ‘1’ for very unimportant, ‘2’ not important, ‘3’ important, ‘4’ very important. The validity test results used Confirmatory Factor Analysis (CFA) of each dimension: (1) Pregnancy (3 items, 0.34–0.88); (2) Childbirth (4 items, 0.38–0.83); (3) Newborn (10 items, 0.61–0.80); and (4) Postpartum (5 items, 0.70–0.79). Their Cronbach alpha’s reliability: 0.69, 0.72, 0.9, and 0.86, respectively. The qualitative research used purposive sampling including 7 village midwives (informant code A) and 6 coordinator midwives (informant code B) in community health centers that used FGD guidelines. 

### 2.3. Research Ethics

Quantitative research data were collected using a Google form questionnaire (online), which was provided via WhatsApp. Detailed information was explained on the initial page of the Google form regarding the researcher’s identity, research title, objectives, content, confidentiality, ethics, questionnaire, length of time to fill out, functions, and benefits of the survey. After the participants read and understood the information, we asked them to voluntarily and independently answer the truth based on their perception towards the questionnaire items. Then, informed consent was filled in (yes or no) upon their agreement. Once agreed, they could go directly to the next page, containing more detailed procedures for answering the questionnaire. Participants received incentives (internet quota) after completing the questionnaire. The research was conducted after obtaining approval from the Health Research Ethics Committee of the Faculty of Medicine, Universitas Padjadjaran, with the number: 640/UN6.KEP/EC/2021.

## 3. Results

### 3.1. Quantitative Research

The results of this study consisted of the characteristics and indicators of competency MCOC on 373 midwife participants in West Java Province, Indonesia. Then, the survey results were explored through qualitative research with a Focus Group Discussion (FGD).

Table 2 shows that midwives who need mobile health-based MCOC competencies have characteristics that mean they were: mostly 35 years old (58.98%); with a working period > 10 years (56.30%); possess a diploma as final education level (72.12%); and the scope of competencies needed by midwives in the early detection and treatment of risk complications in MCOC (84.99%). Most midwives use smartphones 1–12 h/day (78.28%), for social media (28.15%).

Table 3 shows midwives assessed the importance of all indicators of competence in midwifery care, namely three indicators of competence in childbirth, nine for newborns, and one for postpartum care (mean ≥ 3.60). Nevertheless, midwives assessed that the competency indicators for caring for newborns were less critical in mothers with human immunodeficiency virus (HIV), hepatitis, and syphilis (mean < 3.60).

### 3.2. Qualitative Research

Based on the survey results in Table 2 and Table 3, midwives assessed the importance of competence to provide MCOC using an android smartphone, especially for communication, information, and education to patients. Then, qualitative research was conducted through a Focus Group Discussion (FGD) to explore the survey results.

Midwives as informants in qualitative research have the characteristics stated in Table 4, including gender, age, employment status, years of service, last education, and job placement.

The results of the qualitative analysis were explored under three themes including the midwife’s characteristics, MCOC competencies, and health services as the basis for designing mHealth applications (Table 5). The themes involved were: (1) the first theme was the midwife’s characteristics, consisting of three sub-themes: education level, experiences, and professional standards. The increase in the competence of midwives is influenced by their educational background and length of service experiences. Midwives who often handle midwifery cases, means that they gain experience and are motivated to obtain up-to-date information through education and training; (2) the second theme is the MCOC competencies starting from pregnancy, childbirth, newborns, and the puerperium. Midwives need competence in the early detection and treatment of risk complications while providing MCOC based on professional standards and midwife’s authority; (3) the third theme is mHealth. Midwives already use smartphones with various learning applications and health services. In providing MCOC, they need a mobile health application to help record and report the results of continuous maternal and infant health checks. Therefore, the application required a complete MCH services menu such as pregnancy, childbirth, postpartum, infants, and children, telemidwifery containing a combination of digital communication using robots (chatbots) and semi-automatic chatbots (communication with clients using a variety of chatbots and midwives), and inter-professional collaboration (IPC) referral communication with other professions such as a doctor, nutritionist, and health promotion officer. Along with the services menus, education, and training for midwives are essential supports. The image and color of the app’s icon correspond to the MCH book owned by the client for recording the results of MCH checks from when the mother becomes pregnant, gives birth, postpartum, and infant health checks, including immunization.

The themes and sub-themes of competency and service needs in MCOC as the basis for designing mHealth for midwives were developed as learning media for MCH education, training, and health services. The application design includes dashboards, menus, the content of each menu, features, and icon images for menus in mHealth.

Figure 1 shows that midwives providing continuity care need a mHealth application for learning and providing health services including counseling or health education to patients using telemidwifery. By these features, midwives can detect complications early in pregnancy and provide treatment during pregnancy. Thus, it prevents complications in childbirth, newborns, and the puerperium. After providing the services, the midwife’s duties are recording and reporting the results of MCH examinations. The statement was obtained from informants that midwives needed mobile applications because they made it easier to record and report MCH data anytime and anywhere by using a smartphone. 

The mHealth contains a menu based on a cohort form containing data on the health of pregnant women, mothers in labor, infants and toddlers, postpartum, and family planning. Cohort reports have standardized formats from the Ministry of Health. In addition, there is an education menu purposed for midwives and a training menu different from other mHealth applications (details in Figure 2b). The mHealth application also has a telemidwifery menu containing a semi-automatic communication feature between the midwife and the client and chatbot features.

The education menu for midwives guides midwives regarding interventions given to mothers during pregnancy, childbirth, newborns, and postpartum. The mHealth application has a front page shown in Figure 2a,b for users (midwives) who aim to enter and read data. At the same time, the back end (the “kitchen” of the app) shown in Figure 3 and Figure 4 is for the application developers to see a recapitulation of data that the midwife has input.

In Figure 3, midwives can register pregnant, labor, and postpartum mothers’ including toddlers’ identities through the *Posyandu* Reporting Information System for Midwives (PRISM) in the app. Those data will be used with pregnancy, labor, postpartum, and toddler checkup data. Moreover, midwives can make cohort reports, such as pregnancy, baby, toddler, labor, postpartum, and family plan, from stored checkup data. The PRISM also accepts consultations from pregnant women to be submitted to midwives who can provide consulting solutions to pregnant women’s problems. To improve competence, midwives can request educational and training content through PRISM. The content comes from the *Posyandu* Health Content Information System (PHCIS).

The actor of the system is the midwife, who can interact with the eight system services (Figure 4). Midwives can record the results of a toddler, pregnancy, labor, and postpartum checkup. The data obtained from recording these checkups can be processed into a cohort report following the format determined by the government. The telemidwifery service receives consultations from pregnant, labor, and postpartum mothers. Then, to improve their competence, midwives can take advantage of training and education services.

Figure 2a explains that the MCH menu on the mHealth dashboard consists of data on pregnant women, mothers giving birth, infants and toddlers, postpartum, and family planning. Figure 2b shows that the other menus include midwife education about the development of information and health program updates from the government. Moreover, there are standard operating procedures for MCH services, the training menu (which contains an e-module, learning video, and online test), maternal and child health reports called cohorts, telemidwifery, accounts, and log-out features. In addition, the mHealth application can be downloaded on Google Play named the iPosyandu Bidan application (version 1.0.9).

## 4. Discussion 

Midwives need competency in the MCOC health service, including the early detection of the risk factor of complications and treatment management. They are concerned about the purposes, benefits, and design of mHealth to support their needs. In providing services, the midwife’s duties in MCOC are identifying, providing appropriate and safe midwifery care, monitoring, and supporting women during preconception, pregnancy, childbirth, newborn, and postpartum [52]. By strengthening this continuously [1], including in emergencies [53,54], health service quality can be improved [16], preventing complications [4,55]. Midwives should have more quality time when providing MCOC to build relationships with clients, have a sense of community, and respect cultural diversity [56]. In Indonesia, midwives must have competence in MCOC based on the Indonesian Midwifery Constitution No. 4 of 2019, so that midwives carry out the early detection of risk complication cases during pregnancy, childbirth, post-delivery, postpartum, including post-miscarriage care and follow-up with referrals [57]. All women should receive continuous midwifery care as the gold standard for improving maternal and infant health [58]. However, women giving birth who live in a village may have limited access to obtain MCOC due to the characteristics of their population (such as economic, psychological, and social burden on women and their families), the limited number of midwives in the workforce, geography (difficult access to healthcare facilities), and technology (mHealth access) that may affect midwifery care [59]. To support this, the government can support physical and digital infrastructure through which MCOC services can be accessed, including using mHealth [60,61].

Both developed and developing countries should be supported by the competence of midwives to reduce their MCH problems [62]. It is because midwives need the skills, knowledge, and attitudes to provide quality care [63]. A recent review stated that factors influencing the MCH competencies of the midwifery workforce were their educational level (diploma, bachelor, and master degrees), working years, and more education and training experiences [64]. Nonetheless, the midwives’ MCH competencies supporting the MCOC (e.g., cases in detecting the risk of complication and its treatment management in pregnancy, childbirth-related complications, neonatal resuscitation, and management of postpartum hemorrhage) are still inadequate [27,65] because not all midwives meet the cases in their experience in the field [66]. The experience of midwives who work in the field with many cases, including complications, is higher than those with lower numbers of cases [65]. The International Confederation of Midwives (ICM) believes that midwives carrying out competencies must be based on the philosophy of midwifery care because the midwives view the normal birth process in women’s lives from the biological, psychological, and social sides [67]. Midwives also obtain knowledge through critical thinking and experiences that can improve competence after providing MCOC [52]. This competency is needed by midwives to be able to carry out early detection to prevent complications in pregnancy, childbirth, newborns, and the puerperium, as stated in the Decree of the Minister of Health of the Republic Indonesia Number 320 of 2020 [50], and also found in our research (Table 2, Table 3 and Table 4). The assessment of midwives’ training needs is the first step in establishing a training curriculum based on mHealth, that can be used as a national curriculum standard [68]. The assessment is a fundamental aim of professional development in effectively increasing the competence of midwives [69] and for future education and training interventions [25,70]. Education and training programs should be organized according to the needs of midwives based on their service work [71]. Recent reviews stated that practicing midwives lack knowledge of diagnosis and care aspects in the early detection of pre-eclampsia [72], which was also found in our research (Table 3). This early detection is part of midwifery emergency training [73,74]. Education and training for midwives can improve their knowledge and skills to provide better health services [75,76], then reduce maternal and newborn mortality [77].

Technological developments successfully contribute to education and training, midwifery care, and lifelong learning [78]. The effectiveness of technology integration into healthcare training can improve clinical skills professionally and enhance health workers’ learning experiences [79]. Previous studies show that innovative on-traditional methods, such as mHealth, have the potential to increase midwives’ knowledge [72,80]. Besides, educational and skills training that integrates techniques and practice with the mHealth application can increase midwives’ self-efficacy [81] and be a learning medium for training through e-modules and learning videos with blended learning strategies [82,83]. Our research suggests the potential innovation with the combination of midwife competency and midwifery care services (Figure 3 and Figure 4). mHealth, which involves using information and communication technology to implement MCOC [16], can support healthcare to improve the quality of MCH [80,84,85]. It has been publicly utilized and suggests an effective public health service [38]. In our research, mHealth can be used to record maternal health from pregnancy, childbirth, newborn, and postpartum. Midwives can use it for early detection of risks, such as during pregnancy checkups (Figure 2 and Figure 4). When using the app, e.g., for pregnant women, a midwife can detect malnutrition, anemia, and hypertension before determining interventions to prevent complications. 

Previous research stated that mHealth was used for competency applications in patient care, medical knowledge, practice-based learning, its improvement, systems-based practice, and professional, interpersonal, and communication skills. The curriculum with case- and problem-based teaching, supervision, and practice evaluation improves the quality of competencies [63]. mHealth is a promising solution in pregnancy care compared to the standard of maternal care. One of the benefits of mHealth is the media that can be used for digital health communication in telemidwifery [41]. Midwives need telemidwifery to provide health education services for clients (Table 4). Mobile applications can be practical learning tools and significantly transfer information and expertise to midwives [86]. In addition, skills to use mHealth in education and training have become one of the emerging needs for clinical professionals. It will contribute to curricula gaps in the learning process [87]. 

Education and training programs can improve healthcare professionals’ decision-making and communication competencies [69]. A systematic approach, comprising assessment to service, supports the education and training of midwives. Therefore, it helps the recommended need to provide a standard of care following the practice of midwives [88,89]. The programs significantly affect midwives’ leadership [90] and their performance. Thus, competence greatly influences performance appraisal in midwifery services [91]. It is needed to meet public health needs so midwives can provide quality healthcare [35,66]. Midwives working within an organization should be supported to develop their professional roles to become knowledgeable, competent, and confident [92]. The efforts of midwives should be supported by the interoperability between mHealth and the government’s health information system (HIS) [93]. Our research in this iPosyandu for midwives version is in the initial phase and needs more development on the interoperability in the future. It has potential because, previously, the iPosyandu was created to integrate with one of the governmental HIS for *Posyandu* activities, including nutrition status, called ePPGBM [16,38,43]. 

### Strengths and Limitations

The primary strength of the present study is that mHealth combines MCOC competencies and services. In future work, as we are developing chatbots between midwives and mothers, this app will have strengths in telemidwifery [16]. Communicating with a chatbot can help limited numbers of healthcare workers in performing health education services [94]. Our app is planned to be completed with communication features with interprofessional collaboration, such as a doctor and nutritionist. However, it needs more effort to advocate the integration with the governmental midwifery cohort application. Thus, both apps can be synchronized to maintain good data and quality of midwives’ training and services.

## 5. Conclusions

In summary, midwives need competence in MCOC, including the early detection and treatment of risk complications, using mHealth as a learning medium. Amongst its services, mHealth has functions from recording and reporting the results of maternal health examinations to interventions during pregnancy, childbirth, postpartum, and newborn. Education and training programs can improve midwives’ decision-making and communication skills in their healthcare provision. In the future, the development of mHealth will continue with efforts to integrate with the government health information system.

## Figures and Tables

**Figure 1 ijerph-19-13893-f001:**
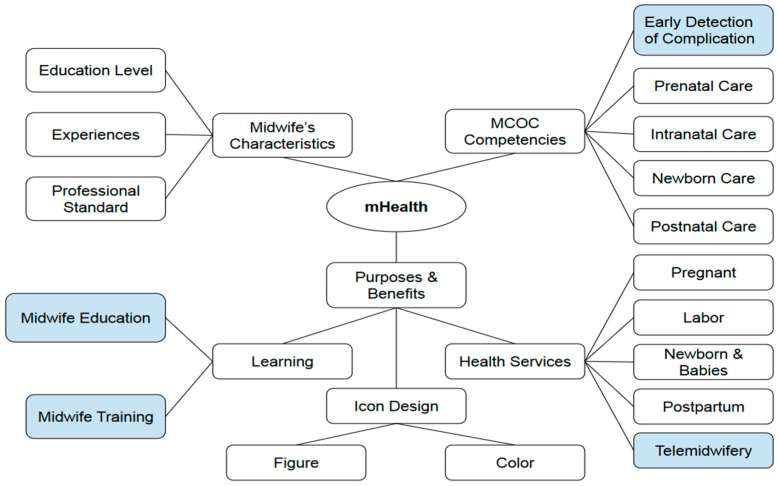
Competence needs in the MCOC mHealth for midwives education (for initial knowledge), training (for strengthening the understanding with skills), and services. Recent findings of qualitative results marked with blue color.

**Figure 2 ijerph-19-13893-f002:**
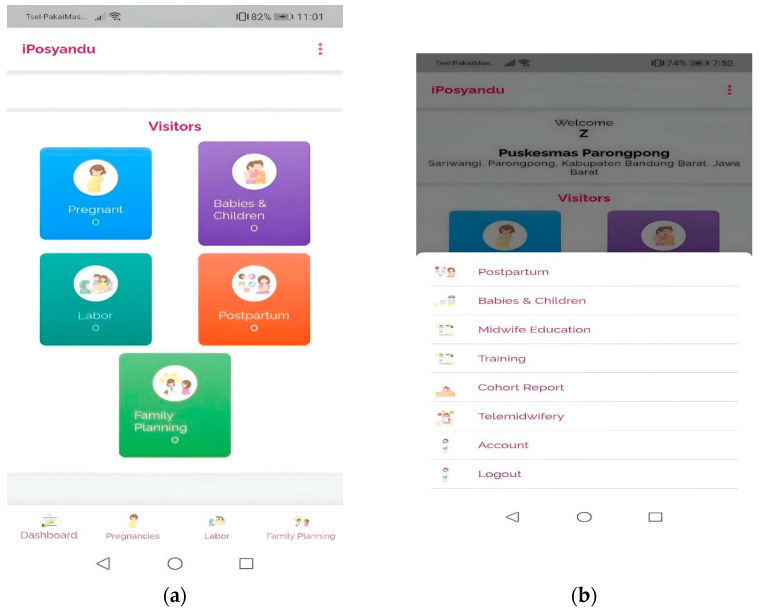
Initiation of mHealth for Midwives: (**a**) Health services menu display; (**b**) Learning menu.

**Figure 3 ijerph-19-13893-f003:**
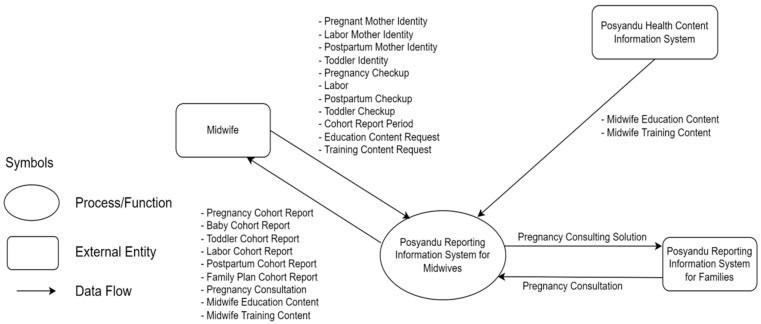
Data Flow Diagram (DFD) level 0 in mHealth application for a midwife.

**Figure 4 ijerph-19-13893-f004:**
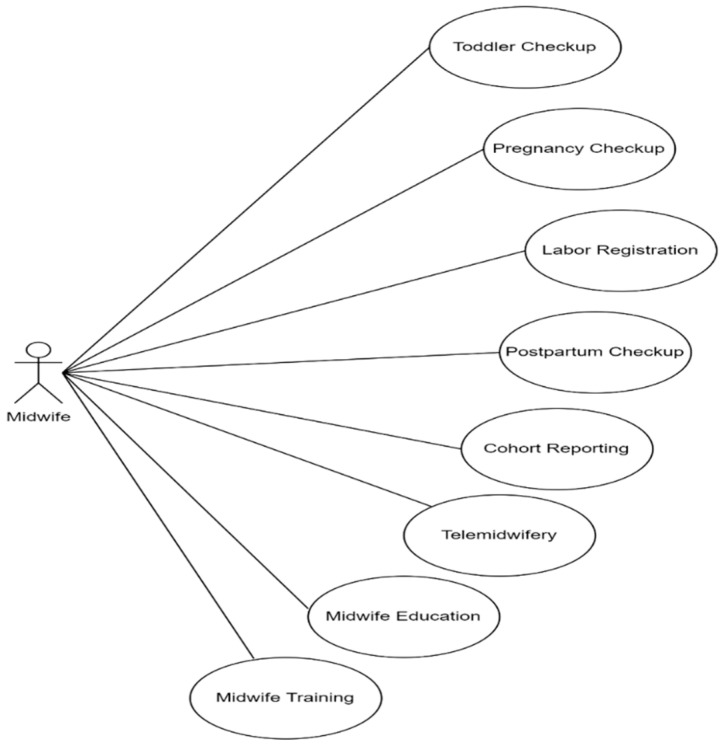
Diagram use case midwife (*bidan*).

**Table 1 ijerph-19-13893-t001:** The sequential explanatory mixed methods design.

Phase	Procedure	Output
**Quantitative**	● Data cross-sectional ● Validity and reliability questionnaire	Descriptive statistics
	STATA 16 data analysis Frequencies Descriptives Univariate	
**Qualitative**	Selection of informants for interviews Develop FGD guideline	Conceptual model of themes Themes and sub-themes
	NVivo Release 1.6.1 data analysis Coding and thematic analysis Within and across-case theme development	
**Explanatory sequential mixed method**	Integration and explanation of quantitative and qualitative findings	Discussion Implication of Practice Future Research

**Table 2 ijerph-19-13893-t002:** Characteristics of midwives.

Characteristics	f	%
Age *		
• ≤35 years	220	58.98
• >35 years	153	41.02
Years of experience		
• <1–5 years	68	18.23
• 6–10 years	95	25.47
• >10 years	210	56.30
Education level		
• Diploma degree (D3)	269	72.12
• Bachelor’s degree (D4/S1)	88	23.59
• Profession	9	2.41
• Master’s degree (S2)	7	1.88
How long does it take to use a smartphone every day?		
• 1–12 h/day	292	78.28
• 13–24 h/day	81	21.72
Smartphones are used for:		
• Learning	35	9.38
• Social media	105	28.15
• Health services	92	24.66
• Others	141	37.80
Scope of Competency MCOC		
• Antenatal Care	20	5.36
• Intranatal Care	31	8.31
• Newborn and Postpartum Care	5	1.34
• Early detection and treatment of risk complications	317	84.99

* age cut-off on adaptation in using mHealth [51].

**Table 3 ijerph-19-13893-t003:** The survey’s results on the continuity of midwifery care competency needs.

No.	Competence	Competency Indicator	Mean ± SD	Min	Max
**1**	Antenatal care/Pregnancy	1. Early detection in pregnancy	3.86 ± 0.35	3.00	4.00
		2. Communication, information, and education on the danger signs of pregnancy	3.82 ± 0.39	3.00	4.00
		3. Counseling for planning delivery and prevention of complications	3.80 ± 0.41	3.00	4.00
**2**	Intranatal Care/Childbirth	4. Early labor screening	3.78 ± 0.41	3.00	4.00
		5. Labor monitoring with partograph	3.85 ± 0.36	3.00	4.00
		6. IV stage of labor monitoring	3.84 ± 0.37	3.00	4.00
		7. Early management of the most common emergency cases in labor	3.74 ± 0.49	1.00	4.00
**3**	Newborn Care	8. Identification of disease problems in newborns	3.65 ± 0.56	1.00	4.00
		9. Identify high-risk babies	3.68 ± 0.53	1.00	4.00
		10. Caring for newborns with human immunodeficiency virus (HIV) mothers	3.42 ± 0.75	1.00	4.00
		11. Care of newborns with hepatitis mothers	3.45 ± 0.80	1.00	4.00
		12. Newborn care with syphilis mother	3.52 ± 0.62	1.00	4.00
		13. Pre-referral baby stabilization	3.67 ± 0.48	2.00	4.00
		14. Early management of premature babies	3.67 ± 0.51	1.00	4.00
		15. Management of resuscitation	3.78 ± 0.46	1.00	4.00
		16. Early management of newborns	3.70 ± 0.48	1.00	4.00
		17. Identify referral needs	3.66 ± 0.49	2.00	4.00
**4**	Postpartum Care	18. Identify problems during the puerperium	3.67 ± 0.49	1.00	4.00
		19. Communication, information, and education about the danger signs of puerperium	3.72 ± 0.47	1.00	4.00
		20. Early management in the puerperium with complications	3.65 ± 0.52	1.00	4.00
		21. Psychosocial support for mothers who have lost their babies	3.67 ± 0.49	2.00	4.00
		22. Early management of emergency cases during the puerperium	3.67 ± 0.49	2.00	4.00

**Table 4 ijerph-19-13893-t004:** Sociodemographic data of the participants of the qualitative study.

Characteristics	Distribution
Gender	13 females
Age	Mean age: (range 28–50 years)
Employment status	Coordinator midwife: 6 Village midwife: 7
Years of service	<7 years: 1 ≥7 years: 12
Last education	Diploma 3 midwifery: 7 Diploma 4 midwifery: 5 Magister public health: 1
Job placement	Urban: 4 Rural: 9

**Table 5 ijerph-19-13893-t005:** Themes and sub-themes competence and service needs in the MCOC education and training mHealth.

Theme	Sub Themes	Quotations
1. Midwife’s Characteristics	i. Education Level	i. *Midwife skills must be continuously improved with training and education* (*Informant B.1*)
	ii. Experiences	ii. *Midwives continue to learn as their experience increases (Informant B.2)*
	iii. Professional standards	iii. *Midwives can sometimes carry out treatment based on authority but are not yet competent (Informant B.3)*
2. MCOC Competencies	i. Early detection and treatment of risk complications	i. *Village midwives carry out the initial handling of complications of pregnancy, childbirth, newborns, and postpartum based on authority but are not yet competent. A referral is made if a case is handled outside the village midwife’s power (Informant B.2)*
	ii. Prenatal Care	ii. *Not all midwives can attend integrated antenatal care training (Informant B.4)*
	iii. Intranatal Care	iii. *Midwives should provide education to patients about preparation for delivery (Informant B.4)*
	iv. Newborn Care	iv. *Need continuous monitoring of newborns until the baby is 28 days old (Informant A.1)*
	v. Postnatal Care	v. *Midwives need to provide education about postpartum repeatedly (Informant B.1)*
3. mHealth	1. Purposes and benefits	i. *So we need an application for continuity care starting from pregnancy, childbirth, or visits to make it easier because the application is on a handphone, so you can take it anywhere to make the midwife’s job easier (Informant A.1)*
		ii. *Continuity midwifery care must be applied, making it easy to find out the mother is in labor (Informant A.4)*
		iii. *Update knowledge with training (Informant B.2)*
		iv. *The dissemination of the training results is shared via the bidan (Informant A.2)*
		v. *Recording and reporting the nutritional status of infants and toddlers using the application (Informant A.3)*
	2. Learning	
	i. Midwife Education	i. *There should be an education menu for midwives in the application so that at least village midwives get initial knowledge before training*. *The training menu should be more experiential than that (the education menu)* *(Informant B.5)*
	ii. Midwife training	ii. *The training menu should be more experiential than that (the education menu)* *(Informant B.5)* iii. *Training materials in the form of modules for theory and videos for skills (Informant A.5)*
	3. Health services	
	i. Pregnant	i. *There is historical data or information for pregnant women*. *(Informant B.6)*
	ii. Labor	ii. *Because it was continuous on the delivery date, there were complications or not, according to the cohort (Informant-A.1)*
	iii. Postpartum	iii. *Used for postpartum maternal health monitoring. (Informant A.1)*
	iv. Babies and children	iv. *Requires baby and toddler nutrition data*. *(Informant A.3)*
	v. Telemidwifery	v. *Communication with doctors, nutritionists, and health promotion the applications from the applications*. *(Informant A.6)* vi. *The continuity of care menu must be sequential, starting from detection and then handling, then up to referral*. *(Informant B.5)* vii. *Prepare answers automatically in the app from frequently asked questions, mom. (Informant A.1)*
	4. Icon	
	i. Figure/image	i. *There are pictures of pregnant women. Then there are pictures of midwives and fathers*. *(Informant A.3)*
	ii. Color application	ii. *The application’s color follows the color of the Mother and Child Health book*. *(Informant A.7)*

## Data Availability

Not applicable.

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
