# Peer review of "Midwifery Continuity of Care in Indonesia: Initiation of Mobile Health Development Integrating Midwives’ Competency and Service Needs"

_ijerph, 2022, doi:10.3390/ijerph192113893_

Round 1

Reviewer 1 Report

This article addresses an interesting topic about the competency needs of midwives in Midwifery Continuity of Care education, training and services using mHealth application. My recommendations are: 

1.Section “Abstract”:

I suggest the inclusion of the time interval of the study. 

2.Section “Materials and Methods”:

2.1. Study design:

·    ·       I suggest the inclusion of the time interval of the study.

·  ·       “…After that, mHealth was developed according to the needs…”  The information about development of a digital service was not mentioned in “Abstract”.

 2.2. Recruitment and participants:

·       ·     “…Quantitative research subjects obtained 373 midwives in …”   The participant number should be mentioned in “Results”.

 ·       The questionnaire was made available on line? This information could be included.

  ·       “The questionnaire comprises 22 items (Table 3)” However, the Table 3 was made available in section “Results”.

 3. Section: “Results”

·       ·       Here, I suggest the inclusion of sub-sections with the results of the quantitative study and other with the results of the qualitative approach.

 ·       The text about Table 2 could be insert before the Table 2.

  ·       Diplom degree in the Table 2: 72.12%. However, in the text: 71.12%. What is the right?

 ·       The text about Table 4 could be insert before the Table 4.

 ·       The authors could insert one table with sociodemographic data of the participants of the qualitative study (13 participants).

 ·       The mHealth developed was not mentioned in “Abstract”

4. Section: “Discussion”

The 1st paragraph could be rewritten, including the summary of the study.

Author Response

Thank you for your suggestions. I have revised the manuscript based on your comments and responses. The detailed responses to your comments will attach in the file as follow:

Reviewer 2 Report

This study provides various insights on the necessity, status, and capacity management of midwives in Indonesia, and is also a study on the current status of midwives management.

In addition, this study is considered to be a good paper that suggests the role and limitations of mobile services for midwives, although there is a limited number of midwives in limited areas.

In order to increase the completeness of the paper, please revise the following.
- Describe the meaning of 'early detection' in more detail in the abstract.
- Explain why the age standard for midwives in the survey was set based on the age of 35.
- In Section 2.3, describe in detail how the midwives who participated in the survey agree to personal information use.
- Also describe how to verify the authenticity of the midwives' answers in the survey.
- Figure 3 (data flow diagram) is too detailed, so it would be good to show it more roughly or delete it. Rather, it would be helpful to understand the mHeath service by presenting more screen captures as shown in Figure 4.
-In Discussion(Chapter 4), write in more detail why midwives are not familiar with MCOC use

Author Response

(The authors gave the same response as above.)

Round 2

Reviewer 1 Report

The authors accepted the suggestions, rewriting the text. However, I ask the authors' attention to the following points:

 POINT 1.

-Abstract: The authors wrote: “This study identifies and explores midwives' competency and service needs to develop mHealth in Midwifery Continuity of Care (MCOC) education and training”

-Introduction: The authors wrote: “This study aims to identify and explore the competency needs of midwives in MCOC education, training, and services using the mHealth application.”

-Results: The authors wrote: “The results of qualitative analysis are used as the basis for designing mHealth applications according to the needs of midwives”

“The mHealth contains a menu based on a cohort form containing data on the health of pregnant women, mothers in labor, infants and toddlers, postpartum, and family planning. Cohort reports have standardized formats from the Ministry of Health. In addition, there is an education menu purposed for midwives and a training menu different from other mHealth applications (details in Figure 4b). The mHealth application also has a telemidwifery menu containing a semi-automatic communication feature between the midwife and the client and chatbot features…”

-Strengths and Limitation: The authors wrote: “Our research in this iPosyandu for midwives version is in the initial phase that needs more development on the interoperability in the future.”

I did not understand. Was the explanatory sequential mixed methods study performed to develop a new mHealth in the context of Midwifery Continuity of Care?  Thus, the authors aim to describe the results of the explanatory sequential mixed methods study and present the new app (Considering these results, a new app was developed).

OR 

Was the explanatory sequential mixed methods study performed to evaluate and adjust one mHealth (iPosyandu) in the context of Midwifery Continuity of Care? Thus, the authors aim to describe these results and present some suggestions for app adjustment. In this case, the authors could describe the current mHealth in the introduction.

The authors could describe more clearly the objectives of the study and rewrite the text according to the context.

POINT 2.

Abstract: “It used an explanatory sequential mixed method, and was conducted from August to 21 December 2021.”

Materials and Methods: “It was conducted from August to December 2021.”

I suggest rewriting the time interval of the study. It is important to describe the date of the quantitative phase and the date of the qualitative approach for a better understanding of the sequence of this study.

POINT 3.

Recruitment and Participants: “Quantitative research subjects obtained 373 midwives participants in West Java Province, Indonesia, using a sampling technique with convenience sampling.”

I suggest writing “Quantitative research subjects included midwives participants in West Java Province, Indonesia, using a sampling technique with convenience sampling”.  

The information “373 midwives” is one result.

POINT 4.

The authors inserted the Table 4 with sociodemographic data of the participants of the qualitative phase. However, there is no mention in the text about Table 4.

Author Response

(The authors gave the same response as above.)
